# Combined Effect of Cinnamon Bark Oil and Packaging Methods on Quality of Fresh Lamb Meat Patties during Storage at 4 °C

**DOI:** 10.3390/foods12152916

**Published:** 2023-07-31

**Authors:** Zubair Hussain, Muawuz Ijaz, Yejun Zhang, Yuqiang Bai, Chengli Hou, Xin Li, Dequan Zhang

**Affiliations:** 1Institute of Food Science and Technology, Chinese Academy of Agricultural Sciences, Key Laboratory of Agro-Products Quality & Safety in Harvest, Storage, Transportation, Management and Control, Ministry of Agriculture and Rural Affairs, Beijing 100193, China; zubairaja530@gmail.com (Z.H.); muawuz.ijaz@uvas.edu.pk (M.I.); bigblue1988@163.com (Y.Z.); yuqiangbai_1844@163.com (Y.B.); houchengli@163.com (C.H.); dequan_zhang0118@126.com (D.Z.); 2Department of Agriculture and Food Technology, Karakorum International University, Main Campus University Road, Gilgit 15100, Pakistan; 3Department of Animal Sciences, CVAS-Jhang 35200, University of Veterinary and Animal Sciences, Lahore 54000, Pakistan

**Keywords:** lamb, meat quality, cinnamon bark oil, MAP

## Abstract

The present study aimed to investigate the effects of adding cinnamon bark oil (CBO) on the quality of ground lamb meat, considering different packaging conditions, including modified atmospheric packaging (MAP) using Hengxian HX-300H and overwrapped packaging. The CBO was incorporated into lamb meat samples at three different levels: 0% (control), 0.025% and 0.05% (*v*/*w*). The samples were then subjected to three packaging methods: MAP1 (80% O_2_ + 20% CO_2_), MAP2 (40% O_2_ + 30% CO_2_ + 30% N_2_) and overwrapped packaging and stored at 4 °C for 0, 4, 8, 12 and 16 days. The findings of the present study revealed that the addition of 0.025% and 0.05% CBO under MAP1 condition significantly improved the color of the meat samples after 12 days of storage at 4 °C (*p* < 0.05). The overwrapped samples exhibited higher levels of thiobarbituric acid reactive substances (TBARS) compared to all other treatments, starting from day 4 of storage (*p* < 0.05). Furthermore, microbial counts were notably higher in the overwrapped samples than in all other samples after day 8 of storage (*p* < 0.05). In conclusion, the combination of 0.05% CBO with MAP proved to be an effective strategy for enhancing the color stability and oxidative stability of ground lamb meat. These results suggest that CBO can be utilized as a beneficial protective agent in meat packaging processes.

## 1. Introduction

The spoilage of meat during storage is a significant concern for the meat industry [1]. Meat is damaged during processing and storage primarily due to microbial growth, lipid and protein oxidation and enzymatic autolysis [2]. These processes lead to deterioration in color and structure [3,4]. Undesirable microbial activity is a major contributor to meat spoilage, which negatively impacts meat quality [5]. This poses a potential threat to meat consumers due to the pathogenicity of microorganisms [6].

Various approaches have been employed to extend the shelf life and improve the quality of meat. Utilizing appropriate packaging and storage conditions can play a crucial role in protecting meat from spoilage caused by intrinsic and extrinsic factors during storage [7]. Recently, MAP methods are used to maintain meat quality and prolong the shelf life of meat. It involves introducing a specific gas atmosphere, which is usually composed of an inert gas like N_2_, O_2_ and CO_2_, to prevent meat microbial growth and color change [8].

Furthermore, the utilization of essential oils (ESOs) in food products combined with various packaging methods proves to be a highly effective strategy for prolonging the shelf life of meat [9,10,11]. These unique ESOs are derived from natural sources and require approval from international and national food authorities to ensure their safety. The cinnamon bark oil has been proven as natural antimicrobial and antioxidant agent, which has been widely employed to preserve food items during storage [12]. Cinnamon bark oil is frequently utilized in food products for its flavor and potential preservative properties. It is commonly employed as a seasoning agent in meat products [13]. Previous studies have demonstrated that essential oils derived from herbs and spices, when used as seasoning in meat products, exhibit varying levels of antimicrobial activity against *Salmonella* spp, *L. monocytogenes*, *E. coli* and *S. aureus* under in vitro conditions [14]. Cinnamon bark oil is a combination of majority (E)-cinnamaldehyde, linalool, β-caryophyllene, eucalyptol and eugenol [15]. Additionally, it has been reported in several studies that EOs including CBO are Generally Recognized as Safe (GRAS) status from the FDA in 21 CFR (Code of Federal Regulation) part 172.515 [16,17]. In our previous experiment, we have used the CBO in ground lamb meat and concluded that cinnamon bark oil of 0.025% and 0.05% had a better preservative effect on the lamb meat quality during storage [12]. Adding CBO with concentrations higher 0.5% negatively affected the color properties during storage [12].

Lamb meat is consumed as a good source of high-quality protein with less fat content and easy digestibility. Many studies have been carried out to know the impact of different packaging techniques on meat quality [12,18,19,20]. However, much less material exists on the impact of ESOs combined with different packaging conditions on the storage life of ground lamb meat patties [21]. Investigating the effects of ESOs and various packaging conditions on lamb meat patties can gain valuable insights into enhancing its storage life and preserving its quality. The current study was conducted to determine the effects of different cinnamon bark oil (CBO) concentrations combined with different MAP conditions on lamb meat samples stored at 4 °C. During storage, different parameters, such as lipid oxidation, microbiological assessment, pH and instrument color attributes, will be studied.

## 2. Materials and Methods

### 2.1. Sample Preparation

Thirty-five lamb carcasses, ranging from 6 to 8 months old with an average body weight of 26.4 ± 3.0 kg, were selected randomly from a local slaughterhouse. Following the standard commercial procedure of the slaughterhouse, all the animals were slaughtered on the same day. Approximately at 24 h’ post-mortem, the *longissimus thoracis et lumborum* (LTL) muscles from the selected lamb carcasses were carefully dissected and transferred to the laboratory, maintaining the refrigeration temperature in an icebox. To ensure cleanliness and preparation for analysis, the visible fat from the surface of the lamb meat samples was meticulously removed. The meat samples were then minced twice, using a 6 mm crushing plate, employing a hygienic mincer machine (FP 3010 Food processor, BRAUA, Frankfurt, Germany). Stringent hygienic conditions were maintained throughout this process to maintain the integrity and quality of the samples.

The CBO (Jiangxi Hengcheng natural flavor oil Co., Ltd., Ji’an, China) concentrations of 0.025% and 0.05% were added to 100 g portions of meat samples in both MAP1 and MAP2. The addition of these concentrations was done manually by carefully mixing them with the meat samples in each tray. Afterward, each prepared sample was placed into a petri dish with a round surface measuring 15 cm in diameter. Five 100-g samples from each carcass from both LTL of a treatment were allocated, i.e., one sample for each time point. After mincing, the samples were assigned into seven treatments (*n* = 5) i.e., overwrapped packaging, MAP1 (0%, 0.025%, 0.05%), MAP2 (0%, 0.025% and 0.05%) packaging samples. The overwrapped packaging was oxygen permeable with oxygen transmission rate of 800 cc/m²/day, moisture vapor transmission rate of 10 g/m²/day and thickness of 25 µm.

For the packaging of MAP, the samples were placed into white polyethylene trays for modified atmospheric packaging with different gas concentrations. The MAP conditions includes MAP1 (80% O_2_ + 20% CO_2_) and MAP2 (40% O_2_ + 30% CO_2_ + 30% N_2_). The food grade gasses with a Hengxian HX-300H (Beijing Hengxian Technology CO., Ltd., Beijing, China) with gas controller Dansensor MAP Mix 9001–3/ 200B (Dansensor, Ringsted, Denmark) was used for producing MAP. The gas atmosphere (O_2_%, CO_2_% and N_2_%) in the packages was checked by Checkmate (Dansensor, Ringsted, Denmark). The white trays with the dimension of 17 cm × 12 cm × 4 cm were used to store the lamb meat samples, and the boxes were heat-sealed using film based on polyamide/polyethylene (LID 550/LID 1050, Sealed Air, Charlotte, NC, USA). For the overwrapped packaging, samples were placed in the white trays laminated with polyvinyl chloride film containing an O_2_ transmission rate of 10,600 cm^3^/ (m^2^·24 h·atm). The packages were immediately stored at the refrigerated temperatures of 4 °C. After the storage days of 0, 4, 8, 12 and 16, the pH, color, myoglobin redox forms, microbial activity and lipid oxidation of the samples were accessed. There were five samples per treatment at each time point.

### 2.2. pH

The pH determination was conducted using a pH meter (Testo 205 pH meter, Lenzkirch, Germany) that was calibrated with standard buffers of pH 4.0 and 7.0 at 25 °C. The pH meter probe was directly immersed into a 1.5 cm thick layer of ground meat, ensuring that it was parallel to the surface. Four measurements were performed for each sample at different locations and then averaged

### 2.3. Color

The color values of the stored meat samples were evaluated following the protocol presented by [12]. The International Commission on Illumination (CIE) parameters, including lightness (L*), redness (a*) and yellowness (b*), were determined for refrigerated samples using a Minolta spectrophotometer CM-600d (Konica Minolta Sensing Inc., Osaka, Japan). The spectrophotometer was equipped with an 8-mm diameter measuring aperture size and set to Illuminant D65 and 10° standard observers. Additionally, the Hue angle and Chroma results were determined using two equations (Equations (1) and (2) provided below). The average color calculations for the lamb meat samples, which had a thickness of 1.5 cm, were randomly estimated from different locations within the sample.

For the calculations on day 0, the results were obtained after mincing and treating the samples under different treatments. All samples’ color values were recorded after being kept for 30 min to bloom at 4 °C for each time point and for all samples. The ratio of reflectance at 630 nm to 580 nm was recorded as a measure of color stability.
Hue angle = tan − 1(b*/a*)(1)
Chroma = (a* 2 + b* 2)1/2(2)

### 2.4. Myoglobin Redox Forms

The myoglobin redox forms were evaluated with specific wavelengths approach [22]. In details, the reflectance intensities at the isobestic wavelengths 474 nm, 525 nm and 572 nm were integrated by the reflectance findings at 470 and 480 nm, 520 and 530 nm, 570 and 580 nm. Furthermore, the reflex attendance results (A) at 474, 525, 572 and 730 nm was estimated with formula A = log10 (1/R). The percentages of the myoglobin redox forms like metmyoglobin (MtMb), deoxymyoglobin (DeoxyMb) and oxymyoglobin (OxyMb) were measured with the help of Equations (3)–(5) as follows:MtMb (%) = [1.395 − (A572 − A730)/(A525 − A730)] × 100(3)
DeoxyMb (%) = 2.375 × [1 − (A474 − A730)/(A525 − A730)] × 100(4)
OxyMb (%) = 100 − DeoxyMb (%) − MetMb (%)(5)

### 2.5. TBARS

The TBARS values were measured to present lipid oxidation by using the method proposed by [23]. Briefly, 10 g of meat was first blended with a sterile homogenizer (SCIENTZ-11, Ningbo Scientz Biotechnology Co., Ltd., Ningbo, Zhejiang, China) containing 25 mL trichloroacitic acid (25% *w*/*w*) and 20 mL distilled water in a sterile homogeneous bag for 2 min. The resulting suspension was again centrifuged for 20 min (3000 rpm), and the supernatant was further employed for TBA test (High-Speed Refrigerated Centrifuge, CR 21N, HITACHI, Tokyo, Japan). The collected supernatant (2 mL) was then mixed with an equal quantity of 0.02 mol/L thiobarbituric acid (TBA; Sigma-Aldrich, St. Louis, MO, USA). The combination was vortexes properly incubated in a boiling water bath for 35 min. Finally, the mixed solution was cooled down to room temperature for absorbance calculations. The absorbance for all samples were assessed at 532 nm using spectrophotometer (Shimadzu UV-1800 Spectrophotometer, Kyoto, Japan), and results were reported as below:TBARS [mg malondialdehyde (MDA)/kg sample] = A532 × 9.48

### 2.6. Microbiological Analysis

For microbial assessment, each sample was analyzed on 0, 4, 8, 12 and 16 days of storage at refrigerated temperature. In details, 10 g of meat samples were aseptically weighed and put into the stomacher bags containing 90 mL (0.1%) saline water followed by agitation for 1 min at 25 °C. After mixing different serial decimal dilution were prepared in 0.1% saline water (Merck, Darmstadt, Germany) for each treatment and triplicate 1 mL sample and then put into selective agar plates. Microbial count was performed in the dishes with microbial load ranging from 30–300 after being incubated for a specific time. The sterile petri dishes were inoculated with serially diluted homogenized samples. After that 15 to 20 mL solution of Plate Count Agar (PCA) was poured and incubated for 48 h (37 °C) [24]. The lactic acid bacteria (LAB) was ascertained on the Man–Rogosa–Sharpe (MRS) medium agar after incubation for 72 h (37 °C) [25]. Serial dilutions were made for the meat samples and inoculated in the petri-dish surface and pored Violet Red Bile Glucose (VRBG) agar solution. After that the petri dishes containing samples and agar solution were incubated at 37 °C for 24 h [26]. Determination of Pseudomonas spp. was made by serial dilutions of meat samples. The meat samples were inoculated in the petri- dish surface and pored 15–20 mL of pseudomonas CFC selective Medium Base solution. After solidifying the petri dishes which containing samples were incubated at 30 °C for 72 h.

### 2.7. Statistical Analysis

Statistical calculations were performed using the SPSS Statistics 21.0 software package (SPSS Inc., Chicago, IL, USA). The results were presented as means with standard errors of the mean (SEM). The collected data was subjected to ANOVA using a mixed model to assess the effects of treatment and storage time. Treatment and storage time were considered fixed factors, while carcasses were treated as a random factor to evaluate the quality of ground lamb meat. The interaction effects of the fixed terms were tested but were found to be non-significant (*p* > 0.05). To determine significant differences among mean values at a significance level of *p* < 0.05, Duncan’s multiple range test was employed as the statistical procedure.

## 3. Results

### 3.1. pH

The pH value was seen among all types of treatment on the ground lamb meat samples stored at 4 °C and presented in Table 1. After the storage for 16 days at 4 °C, the pH values of overwrapped packaged samples showed a significant (*p* < 0.05) decrease on day 16 compared to the CBO treated samples. However, no variation in the pH values were observed for all stored samples on day 0.

### 3.2. Color

The instrumental color values (L*, a*, b*, Chroma, Hue) of different treatments on lamb meat samples during refrigerated storage were presented in Table 2. The L* value of 0.05% CBO treatment samples under MAP1 was notably (*p* < 0.05) greater on day 8 than that of 0% MAP1 and Ow packed samples. However, the treatment samples 0.025% and 0.05% of CBO with MAP2 was remarkably (*p* < 0.05) greater on day 8 as compared to 0% samples stored under MAP2 and overwrapped packaged samples. On days 12 and 16 the L* values treatment samples 0.025% and 0.05% of CBO under MAP1 were remarkably higher than all other samples while, on days 12 and 16, no variation was seen among MAP2 treated samples.

The a* value of stored treatments 0.025% and 0.05% of CBO under MAP1 condition was considerably greater on day 4 compared to 0% MAP1, MAP2 samples and Ow packed samples. On day 8, the a* value was notably higher in 0.025% CBO under MAP1 than those of all other samples (*p* < 0.05). No variation was seen among MAP2 stored samples on day 8. On days 12 and 16 the samples treated under MAP1 were considerably (*p* < 0.05) higher than overwrapped samples with no significant results among MAP1 stored samples. Overall, it was noticed that the treatments 0.025% and 0.05% of CBO under MAP1 storage condition retained a good redness value after day 12 during the storage than other stored samples.

The b* value in the samples stored with 0.05% CBO under MAP2 condition was considerably greater on day 4 than other treatments except 0% under MAP2 and overwrapped packaged treatments (*p* < 0.05). On day 8, no variation was seen among the stored treatments under MAP1 condition. However, the b* in the 0% samples under MAP2 condition was significantly higher on day 8 than all other treated samples except overwrapped packaged samples and 0.05% CBO meat samples under MAP2 condition (*p* < 0.05). On day 12, the b* value was significantly lower in 0.05% CBO samples under MAP1 as compared to other samples. No significant difference was observed in 0% samples under MAP1 condition (*p* < 0.05). The b* value of treatments under overwrapped and 0%, 0.025% CBO under MAP2 treatments on day 16 was significantly higher as compared to 0.025% and 0.05% CBO under MAP1condition (*p* < 0.05).

On day 4 there was no significance difference among samples stored under MAP1. The Chroma value was lower in 0.025% CBO samples stored under MAP2 condition on day 4 as compared to all other samples (*p* < 0.05). On day 8, higher Chroma value was noted higher in 0.025% of CBO samples stored under MAP1 condition in comparison with other MAP1 treatments (*p* < 0.05). The Chroma value in 0% samples stored under MAP1 had significantly (*p* < 0.05) higher value on day 12 than overwrapped packaged samples and 0% sample stored under MAP2 condition. At the end of storage days, the treatment samples MAP1 and MAP2 had significantly greater Chroma values in comparison with overwrapped packaged samples (*p* < 0.05).

The Hue angle of meat samples treated 0.05% CBO under MAP2 condition was remarkably greater on day 4 than 0.025% CBO under MAP2 condition and all other samples (*p* < 0.05). Whereas, no significance was among stored samples of MAP1. On day 8, the stored sample 0% under MAP2 condition was significantly (*p* < 0.05) higher than other samples. No difference was observed with 0.05% CBO under MAP2 condition. The Hue angle of overwrapped and MAP2 stored samples were significantly greater on day 12 as compared to MAP1 stored samples (*p* < 0.05). On day 16, the Hue angle in overwrapped packaged samples was significantly greater than all other samples.

### 3.3. Relative Contents of Myoglobin Redox Forms

The results of myoglobin redox forms in ground lamb meat samples stored at 4 °C are presented in Table 3. No significant differences were observed among the OxyMb, DeoxyMb and MetMb levels on day 0 of storage. Throughout the storage period, the DeoxyMb was significantly higher in the overwrapped samples compared to the other treatments (*p* < 0.05). On day 16, the samples stored under MAP1 conditions with 0.025% and 0.05% CBO demonstrated significantly lower levels than all other samples (*p* < 0.05). Conversely, the OxyMb were considerably higher in the 0.025% and 0.05% CBO treatments stored under MAP1 from days 12 to 16, in comparison with the other samples. The MetMb showed significant increase in the overwrapped packaged samples, reaching the consumer acceptability limit of 40%. Whereas, all samples stored under MAP2 conditions showed an increasing trend after day 8 compared to the samples stored in MAP1 storage condition (*p* < 0.05). The MetMb values in the overwrapped packaged samples were significantly higher from days 8 to 16, compared to the other treatments. The MetMb values of samples with 0.025% and 0.05% CBO stored under MAP1 were remarkably lower on days 12 and 16 than MAP2 treatments (*p* < 0.05).

### 3.4. TBARS

The increase of TBARS levels in the ground lamb meat treated with and without CBO packed under overwrapped, MAP1 and MAP2 at 4 °C for 16 days are presented in Table 4. The results show the addition of TBARS in the stored samples, with the levels not exceeding 1.16 mg MDA/kg in both MAP1 and MAP2 packages throughout the entire 16 days’ storage period. Interestingly, the TBARS level in the overwrapped packed samples was significantly higher between days 8 and 16 compared to all other formulated samples, reaching 2.26 mg MDA/kg (*p* < 0.05). On the other hand, the formulated meat samples containing 0.025% and 0.05% CBO and stored under MAP2 conditions exhibited significantly lower TBARS values on days 12 and 16 compared to MAP1 samples (*p* < 0.05). No significant difference was observed between 0% CBO samples stored under MAP2 condition. In general, it was evident that the combination of CBO with both MAP packaging conditions had the ability to retard the formation of TBARS during the storage.

### 3.5. Microbiological Properties

The variation within the populations of TVC in the formulated lamb meat samples during 4 °C storage was illustrated in Figure 1a. On the first day, the TVC populations in all treatment samples ranged from 3.4 to 3.52 log CFU/g. The addition of CBO in both MAP1 and MAP2 resulted in a reduction in microbial growth compared to the overwrapped samples during the storage (*p* < 0.05). There was no significant difference observed on days 0 and 4 among the treatment groups during the storage period. The TVC in overwrapped packed samples showed notably higher counts on days 8, 12 and 16 compared to all other treatments groups, exceeding 7 log CFU/g (*p* < 0.001). In comparison with other samples, the TVC were significantly lower in the samples stored under 0.025% and 0.05% CBO under MAP2 condition on day 16, measuring 5.43 and 5.35 log CFU/g, respectively.

The fluctuations in the populations of enterobacteriaceae counts during the storage up to 16 days were illustrated in Figure 1b. Initially, there was no significance difference in the enterobacteriaceae between the treatments which ranged from 3.45–3.54 log CFU/g. The populations of enterobacteriaceae rapidly increased in the overwrapped samples reaching 7.5 log CFU/g on day 16 than other treatments (*p* < 0.05). On the other hand, the populations of enterobacteriaceae within CBO stored samples under MAP1and MAP2 packaging conditions were notably increased after days 8 to 16 during storage. The enterobacteriaceae counts were reduced by CBO treatments 0.025% and 0.05% under MAP2 condition on day 16 which was 5.25 and 5.34 log CFU/g (*p* < 0.05).

The increasing trend of lactic acid bacteria were noticed in the overwrapped packed samples ranging from 3.54–6.59 log CFU/g while retarded by adding CBO in the meat samples as demonstrated in Figure 1c. The inclusion of CBO samples in both MAP1 and MAP2 conditions effectively slowed down the growth of lactic acid bacteria during the storage (*p* < 0.05). The lactic acid bacteria in overwrapped packed samples were remarkably greater on days 8 and 12 compared to other treatments (*p* < 0.05). No significant difference was noticed in both MAP treatment samples. On day 16, the population of lactic acid bacteria in overwrapped samples was significantly higher than all other stored samples. The CBO stored samples 0.025% and 0.05% under MAP2 condition recorded a range of 5.34–5.28 log CFU/g on day 16, which was significantly lower than those of both MAP1 and MAP2 stored samples (*p* < 0.05).

The pseudomonas counts in the samples stored without CBO treatments exhibited an increasing trend as depicted in Figure 1d. Initially, there were no significance differences in pseudomonas counts between the treatments, ranging from 3.46–3.59 log CFU/g. However, on days 12 and 16, the pseudomonas counts in overwrapped samples were considerably higher measuring 6.19–7.15 log CFU/g, compared to all other treatments (*p* < 0.05). Both MAP1 and MAP2 conditions, with or without CBO, showed an increasing trend in pseudomonas counts from days 8 to 16 during storage (*p* < 0.05).

## 4. Discussion

The current study investigated the positive effects of incorporating CBO along with various packaging methods on meat quality parameters. A preliminary experiment was conducted on ground lamb meat using different concentrations of cinnamon bark oil (CBO) and MAP gasses. It was observed that the addition of cinnamon bark oil at concentrations higher than 0.5% effectively inhibited the lipid oxidation and the proliferation of microorganisms in the ground lamb meat samples. It had a negative impact on the color and odor of the meat. Our previous study identified concentrations of 0.025% and 0.05% as the most effective under vacuum packaging [12]. Similarly, selected packaging conditions were found to be more effective than other methods and were further used for the current experiment.

The results indicated that the pH value of overwrapped samples was higher on day 16 compared to the other samples. This increase in pH value can be attributed to the breakdown of protein molecules, leading to the production of chemicals and ammonia. These compounds can potentially promote the growth of pathogenic microorganisms, as suggested by previous studies [27,28].

Another crucial aspect of meat quality is its color, which can significantly influence customer acceptability [29]. In this study, the color properties (L*, a*, b*, Chroma and Hue angle) of ground lamb meat, packed under different packaging conditions in combination with CBO, were observed to change over storage time. These alterations in color characteristics may have implications for the overall quality of the meat and its desirability to consumers [30]. According to [31], for customer acceptability, the threshold levels for the color of fresh lamb meat displayed should be above 14.5 for a* and 44 for L*. In our study, the L* values of the CBO samples stored under MAP1 at concentrations of 0.025% and 0.05% were significantly higher (*p* < 0.05) after day 12 than those of the other stored samples, exceeding the threshold level of 44 during storage. [20] reported that the lightness values increased in muscle with the duration of storage for samples stored under either 20% CO_2_ + 10% O_2_ + 70% N_2_ or 80% CO_2_ + 20% O_2_, ranging from 45.32 to 48.92. The rise in the L* value may be attributed to increased light scattering, protein degradation and pH changes in the muscles [32,33,34].

The measurement of the a* value in meat serves as a significant indicator for assessing meat color and its impact on consumer acceptability [30], particularly in lamb. As shown in Table 2, the a* values of CBO samples stored under MAP1 at concentrations of 0.025% and 0.05% were higher than those stored under MAP2 and samples stored with overwrap packaging. Furthermore, the a* values of 0.025% CBO samples stored under MAP1 exceeded the threshold value of 14.5, which is suggested for higher consumer acceptance according to [31]. It is worth noting that a higher proportion of oxygen in the modified atmosphere conditions is responsible for the bright red color observed on the meat surface, as reported by [35]. Previous studies have demonstrated that a higher concentration of oxygen in the MAP is more favorable for enhancing meat redness (higher a* value and lower hue angle) compared to vacuum-packed pork samples [33]. The presence of higher oxygen levels in the packaging may prolong the development of MetMb (metmyoglobin) formation, which is responsible for meat spoilage, as indicated by [36]. The b* values were higher in overwrapped samples and in stored samples treated with 0% and 0.025% CBO stored under MAP2 on day 16 compared to other samples. The elevated b* parameter could be attributed to the aggregation of MetMb on the surface of stored samples, which have lower levels of oxygen concentration. The assessment of myoglobin forms is crucial for determining color degradation [31]. In this study, MetMb formation was significantly higher in overwrapped samples, followed by MAP2 and MAP1. The concentrations 0.025% and 0.05% CBO stored under MAP1 were the most effective for inhibiting the oxidation of myoglobin as compared to other stored samples. The lower oxidation of myoglobin could be attributed to the antioxidant activity of bioactive compounds found in the CBO. The compounds possess the capability to scavenge and neutralize free radicals, which are formed during the oxidation process. The conversion of oxymyoglobin to MetMb hindered the redness of meat samples, leading to color spoilage [37]. The higher amount of MetMb in the meat samples may also intensify the level of lipid oxidation [38].

During storage, all meat samples maintained the level of lipid oxidation below the acceptable limit (<2 mg MDA/kg) except overwrapped packaging on day 16 [39]. The initial TBARS (thiobarbituric acid reactive substances) levels in the meat samples were not significantly different. As the storage days increased, there was a noticeable upward trend in TBARS levels. This increase can be attributed to the oxidative conditions experienced by the samples during the storage period. After 16 days of storage, the TBARS levels were found to be highest in the overwrapped meat samples, followed by MAP1 and MAP2. The reduction in the TBARS levels in the MAP meat samples might be due to the various phenolic compounds present in CBO, including cinnamaldehyde, which possesses antioxidant properties. The phenolic compounds in CBO have the ability to scavenge free radicals, which are highly reactive molecules that contribute to lipid oxidation. By neutralizing these free radicals, CBO can help to prevent or delay lipid oxidation in the meat [40]. The presence of a lower oxygen atmosphere has been reported to slow down the formation of TBARS [41]. Higher TBARS levels were observed in the meat samples stored under a higher level of oxygen [42]. Furthermore, [43] confirmed that a mixture of 75% oxygen and 25% carbon dioxide in the atmospheric treatment resulted in a higher degree of oxidation in packaged beef steaks. The addition of astaxanthin at the levels ranging from 20 to 80 mg/kg in raw and cooked lamb patties has been found to effectively retard the lipid oxidation during refrigerated storage, thus enhancing the shelf life of lamb meat patties [44]. In another study, it was found that the combination of rosemary essential oil along with modified atmospheric packaging demonstrated a robust preservative effect, effectively prolonging the shelf life of meat [9]. Similarly, it was noticed that the meat product shelf life can be extended by introducing the natural antioxidant 0.1% thyme oil under MAP (80% CO_2_/20% N_2_) [10]. Furthermore, nutraceuticals, such as resveratrol and citroflavan-3-ol, retard the lipid oxidation in raw lamb meat patties and can prove useful as safe, natural and health promoting antioxidants to the meat industry [21].

The microbial activities increased in all treatments as the storage days increased. In the current study, the growth rate of TVC and enterobacteriaceae crossed 7 log CFU/g in the samples stored under overwrapped packaging. The International Commission on Microbiological Specifications for Foods [45,46] has established a threshold level for the total microbial count in meat cuts. If the microbial population is higher than 7 log CFU/g, it can result in noticeable off-odor, sourness and visible slime formation [47]. The higher microbial contamination could be attributed to the atmospheric conditions within the packages of minced beef patties [48]. It was noticed in the previous studies that the microbial growth is inhibited by a higher concentration of CO_2_ during packaging [49].

The populations of TVC (total viable count) and enterobacteriaceae in the 0% stored samples under MAP1 were significantly higher (*p* < 0.05) after days 8 to 16 compared to the 0.05% CBO stored samples under MAP2 during storage. Overall, it was noticed that the 0.05% CBO stored samples under MAP2 significantly inhibited the activities of microorganism compared to control samples during storage. The increased microbial activity in the 0% stored samples under MAP1 can be due to the absence of CBO and lower levels of CO_2_. Previously, it was confirmed that the CBO contains natural active compounds, such as cinnamaldehyde, which can effectively inhibit microbial growth in stored food products [50]. The active compounds inhibit bacteria by damaging cell membrane; altering the lipid profile; inhibiting ATPases, cell division, membrane porins, motility, and biofilm formation [51]. In a study by [52], it was reported that higher levels of CO_2_ in packaging can hinder microbial proliferation in meat samples during storage. The active compounds present in CBO are responsible for disrupting the cell wall, cytoplasmic membrane and genetic information, as demonstrated by [53]. It is important to mention that sensorial properties of Chinese cinnamon and cinnamon bark oil at the level of 0.05% (*v*/*v*) was acceptable in terms of smell and taste in ready to cook ground meat [54].

## 5. Conclusions

Adding CBO could effectively retard the microbial growth in both MAP conditions. The samples stored under MAP2 condition with 0.05% CBO had a better protective effect on inhibiting the populations of TVC, lactic acid bacteria, enterobacteriaceae and pseudomonas during storage. CBO could be used as an efficient preservative agent in combination with MAP to improve the color stability and oxidative stability of ground lamb meat.

## Figures and Tables

**Figure 1 foods-12-02916-f001:**
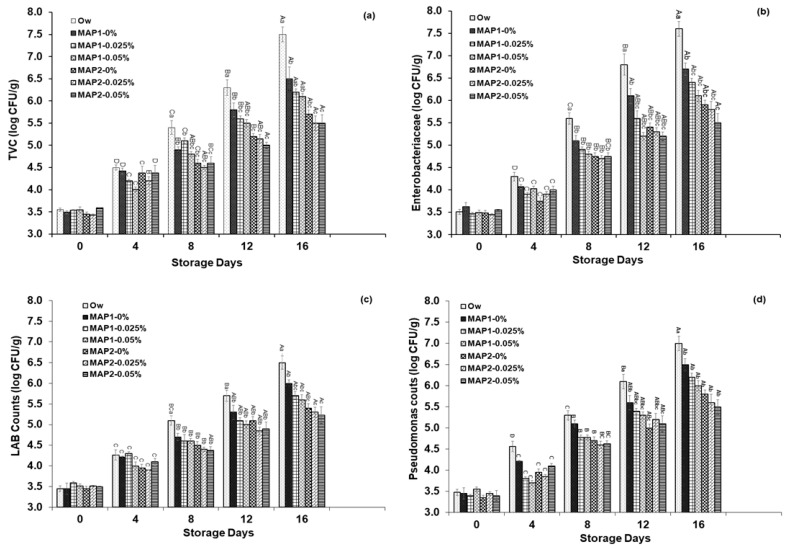
Effect of cinnamon bark oil on the total viable counts (**a**), enterobacteriaceae (**b**), lactic acid bacteria (**c**) and pseudomonas (**d**) populations in ground lamb meat under overwrapped (Ow), MAP1 (80% O_2_ + 20% CO_2_) and MAP2 (40% O_2_ + 30% CO_2_ + 30% N_2_) storage at 4 °C. Different markers show the mean values of different treatments while the bars indicate standard error at each sampling point. ^A–D^ Mean with different letters indicated that changes during storage differ significantly (*p* < 0.05), ^a–c^ Mean with different letters showed that changes between treatments differ significantly (*p* < 0.05).

**Table 1 foods-12-02916-t001:** Effects of adding cinnamon bark oil on the pH of ground lamb stored at 4 °C for 16 days under different packaging conditions (*n* = 5).

Treatments		Storage Time (Day)						
Packaging Types	CBO	0	4	8	12	16	SEM	*p*-Value
Ow	0%	5.43 ^d^	5.50 ^Dc^	5.66 ^ABb^	5.70 ^Ab^	5.92 ^Aa^	0.07	<0.002
MAP1	0%	5.49 ^d^	5.60 ^BCc^	5.59 ^Cc^	5.68 ^Ab^	5.78 ^Ba^	0.07	<0.001
	0.025%	5.46 ^bc^	5.57 ^CDb^	5.69 ^Aa^	5.70 ^Aa^	5.68 ^Ca^	0.09	<0.001
	0.05%	5.50 ^c^	5.64 ^ABb^	5.62 ^BCb^	5.68 ^Ab^	5.77 ^Ba^	0.10	<0.001
MAP2	0%	5.46 ^c^	5.67 ^Ab^	5.63 ^BCb^	5.69 ^Ab^	5.78 ^Ba^	0.11	<0.001
	0.025%	5.54 ^bc^	5.55 ^CDc^	5.65 ^ABb^	5.71 ^Ab^	5.81 ^Ba^	0.10	<0.001
	0.05%	5.47 ^b^	5.58 ^Cb^	5.71 ^Aba^	5.69 ^Aa^	5.74 ^BCa^	0.09	<0.001
*p*-value		<0.59	<0.001	<0.001	<0.336	<0.001		

^A–D^ Means in the same column with different superscript of capital letters differ significantly (*p* < 0.05); ^a–d^ Means in the same row with different superscript of small letters differ significantly (*p* < 0.05). Ow: overwrapped, MAP1: 80% O_2_ + 20% CO_2_, MAP2: 40% O_2_ + 30% CO_2_ + 30% N_2_.

**Table 2 foods-12-02916-t002:** Effect of adding cinnamon bark oil on the color measurements of ground lamb stored at 4 °C for 16 days under different packaging conditions (*n* = 5).

	Treatments		Storage Time (Day)						
	Packaging Types	CBO	0	4	8	12	16	SEM	*p*-Value
L*	Ow	0%	41.6 ^b^	43.0 ^Ca^	40.1 ^Db^	41.6 ^Db^	39.0 ^Cc^	0.20	<0.005
	MAP1	0%	41.4 ^c^	46.3 ^ABa^	43.7 ^Bb^	47.3 ^ABa^	47.1 ^Ba^	0.40	<0.001
		0.025%	42.6 ^c^	47.9 ^Aab^	45.7 ^ABb^	48.1 ^Aab^	48.8 ^Aa^	0.23	<0.001
		0.05%	40.7 ^c^	46.9 ^Ab^	47.1 ^Ab^	48.3 ^Aab^	49.2 ^Aa^	0.29	<0.001
	MAP2	0%	40.8 ^b^	44.8 ^BCa^	44.8 ^Ba^	44.8 ^Ca^	44.6 ^BCa^	0.29	<0.001
		0.025%	41.0 ^c^	46.5 ^ABa^	46.7 ^Aa^	45.1 ^Cb^	46.0 ^Ba^	0.35	<0.001
		0.05%	42.2 ^b^	46.1 ^ABa^	46.8 ^Aa^	46.1 ^BCa^	46.8 ^Ba^	0.37	<0.001
	*p*-value		<0.290	<0.001	<0.001	<0.001	<0.001		
a*	Ow	0%	12.1 ^a^	12.3 ^BCa^	12.5 ^Ca^	9.8 ^Cb^	7.9 ^Cc^	0.12	<0.001
	MAP1	0%	12.3 ^b^	13.0 ^Bb^	13.8 ^Bb^	14.8 ^Aa^	13.8 ^ABb^	0.44	<0.001
		0.025%	12.1 ^b^	14.6 ^Aa^	15.9 ^Aa^	14.2 ^Aa^	14.5 ^Aa^	0.20	<0.001
		0.05%	13.0 ^b^	14.5 ^Aa^	13.9 ^Bab^	13.2 ^Aab^	14.9 ^Aa^	0.22	<0.108
	MAP2	0%	12.1 ^a^	11.2 ^Ca^	11.3 ^Da^	11.4 ^Ba^	13.0 ^Ba^	0.38	<0.341
		0.025%	12.8 ^a^	11.5 ^Cb^	12.3 ^CDa^	10.7 ^BCb^	12.3 ^Ba^	0.21	<0.056
		0.05%	11.9 ^a^	11.6 ^Ca^	11.8 ^CDa^	10.0 ^BCb^	12.8 ^BCa^	0.25	<0.243
	*p*-value		<0.266	<0.0001	<0.001	<0.001	<0.001		
b*	Ow	0%	11.7 ^b^	13.9 ^ABa^	15.3 ^ABCa^	15.7 ^ABa^	16.3 ^Aa^	0.19	<0.001
	MAP1	0%	11.9 ^c^	13.2 ^Bb^	13.0 ^Db^	14.1 ^BCa^	15.4 ^ABa^	0.19	<0.000
		0.025%	11.9 ^c^	12.5 ^Bbc^	14.1 ^BCa^	14.8 ^ABa^	13.9 ^Bab^	0.25	<0.027
		0.05%	11.1 ^c^	13.6 ^Bab^	12.8 ^Db^	13.6 ^Cab^	14.7 ^Ba^	0.23	<0.007
	MAP2	0%	11.8 ^d^	14.3 ^ABc^	15.9 ^Aab^	14.9 ^ABbc^	16.7 ^Aa^	0.17	<0.001
		0.025%	11.2 ^c^	12.3 ^Bc^	14.2 ^Cb^	15.1 ^ABab^	16.4 ^Aa^	0.24	<0.001
		0.05%	11.7 ^b^	15.6 ^Aa^	15.7 ^ABa^	16.2 ^Aa^	15.6 ^ABa^	0.18	<0.001
	*p*-value		<0.697	<0.014	<0.001	<0.004	<0.001		
Chroma	Ow	0%	16.8 ^c^	18.6 ^ABb^	19.0 ^Ba^	17.8 ^Cc^	17.4 ^Bb^	0.18	<0.001
	MAP1	0%	17.1 ^b^	18.4 ^ABab^	18.3 ^Bab^	20.7 ^Aa^	20.5 ^Aa^	0.33	<0.023
		0.025%	17.0 ^c^	19.0 ^Ab^	21.2 ^Aa^	19.4 ^ABb^	20.5 ^Aab^	0.28	<0.001
		0.05%	17.4 ^b^	19.9 ^Aa^	18.1 ^Bab^	18.7 ^ABab^	19.8 ^ABa^	0.32	<0.057
	MAP2	0%	16.9 ^c^	18.2 ^ABbc^	19.6 ^Bab^	18.1 ^BCbc^	21.5 ^Aa^	0.38	<0.006
		0.025%	17.0 ^b^	17.1 ^Cb^	19.0 ^Bab^	18.5 ^ABab^	21.5 ^Aa^	0.29	<0.001
		0.05%	16.6 ^b^	19.5 ^Aa^	19.7 ^Ba^	19.9 ^ABa^	19.7 ^ABa^	0.20	<0.001
	*p*-value		<0.882	<0.118	<0.002	<0.039	<0.070		
Hue	Ow	0%	43.9 ^d^	48.9 ^BCc^	50.8 ^Bc^	56.3 ^Ab^	63.6 ^Aa^	0.14	<0.001
	MAP1	0%	44.1 ^a^	44.9 ^CDa^	44.9 ^Ca^	45.0 ^Ba^	48.1 ^CDa^	0.84	<0.001
		0.025%	44.6 ^a^	41.0 ^Da^	44.5 ^Ca^	44.1 ^Ba^	43.5 ^DEa^	0.65	<0.662
		0.05%	41.7 ^a^	42.4 ^Da^	41.6 ^Ca^	45.8 ^Ba^	41.6 ^Ea^	0.60	<0.177
	MAP2	0%	44.5 ^b^	51.9 ^ABa^	54.8 ^Aa^	52.1 ^Aa^	51.9 ^BCa^	0.65	<0.001
		0.025%	41.5 ^c^	47.8 ^Cb^	49.8 ^Bb^	54.7 ^Aa^	51.3 ^BCa^	0.49	<0.001
		0.05%	44.5 ^b^	53.3 ^Aa^	53.2 ^ABa^	56.4 ^Aa^	53.3 ^Ba^	0.73	<0.001
	*p*-value		<0.985	<0.001	<0.001	<0.001	<0.001		

^A–E^ Means in the same column with different superscript of capital letters differ significantly (*p* < 0.05); ^a–d^ Means in the same row with different superscript of small letters differ significantly (*p* < 0.05). Ow: overwrapped, MAP1: 80% O_2_ + 20% CO_2_, MAP2: 40% O_2_ + 30% CO_2_ + 30% N_2_.

**Table 3 foods-12-02916-t003:** Effect of adding cinnamon bark oil on the metmyoglobin redox forms of ground lamb stored at 4 °C for 16 days under different packaging conditions (n = 5).

	Treatments		Storage Time (Day)						
	Packaging Types	CBO	0	4	8	12	16	SEM	*p*-Value
DeoxyMb	Ow	0%	51.3 ^a^	25.4 ^Acd^	30.3 ^Ab^	27.6 ^Ac^	24.3 ^Ad^	0.20	<0.001
	MAP1	0%	52.3 ^a^	16.2 ^Bc^	18.3 ^Bc^	20.3 ^Bb^	17.5 ^BCc^	0.40	<0.021
		0.025%	53.3 ^a^	15.1 ^BCb^	16.6 ^BCb^	14.6 ^Cb^	11.6 ^Dc^	0.23	<0.001
		0.05%	52.3 ^a^	14.6 ^BCb^	13.6 ^Cb^	12.9 ^Db^	13.6 ^Db^	0.29	<0.001
	MAP2	0%	51.3 ^a^	14.6 ^BCd^	16.2 ^BCcd^	17.0 ^BCbc^	19.1 ^Bb^	0.29	<0.001
		0.025%	52.2 ^a^	13.1 ^Cc^	15.1 ^BCbc^	16.1 ^BCb^	17.0 ^BCb^	0.35	<0.001
		0.05%	51.2 ^a^	14.1 ^Bbc^	15.1 ^BCb^	13.1 ^CDb^	15.1 ^Cb^	0.37	<0.001
	*p*-value		<0.291	<0.001	<0.001	<0.001	<0.001		
OxyMb	Ow	0%	26.2 ^b^	36.5 ^Ca^	26.3 ^Cb^	24.5 ^Fb^	19.0 ^Dc^	0.19	<0.001
	MAP1	0%	26.2 ^d^	60.4 ^Aa^	56.2 ^Bb^	50.8 ^Dc^	49.6 ^Bc^	0.46	<0.001
		0.025%	26.2 ^d^	63.3 ^Aa^	60.5 ^Ab^	58.5 ^Bc^	57.9 ^Ac^	0.25	<0.001
		0.05%	26.2 ^c^	62.9 ^Aa^	61.8 ^Aa^	59.5 ^Aab^	54.8 ^Ab^	0.45	<0.001
	MAP2	0%	27.5 ^e^	56.5 ^Ba^	55.2 ^Bb^	47.1 ^Ec^	33.9 ^Cd^	0.38	<0.001
		0.025%	26.2 ^e^	62.3 ^Aa^	57.9 ^ABb^	50.3 ^Dc^	39.4 ^Cd^	0.26	<0.001
		0.05%	27.2 ^d^	60.3 ^Aa^	55.3 ^Bb^	55.0 ^Cb^	42.9 ^Bc^	0.45	<0.001
	*p*-value		<0.766	<0.001	<0.001	<0.001	<0.001		
MetMb	Ow	0%	19.3 ^e^	26.7 ^Ad^	32.5 ^Ac^	42.5 ^Ab^	55.3 ^Aa^	0.19	<0.001
	MAP1	0%	18.5 ^d^	20.6 ^Bc^	25.7 ^Fb^	29.1 ^Db^	33.0 ^Ea^	0.19	<0.001
		0.025%	18.9 ^d^	21.7 ^Bc^	23.9 ^EFc^	27.1 ^Eb^	30.7 ^Ea^	0.25	<0.001
		0.05%	19.9 ^e^	22.6 ^Bd^	24.7 ^Dc^	27.7 ^Eb^	31.6 ^Da^	0.23	<0.001
	MAP2	0%	20.9 ^e^	24.0 ^Ad^	28.7 ^BCc^	35.9 ^Bb^	46.9 ^Ba^	0.17	<0.001
		0.025%	20.7 ^e^	24.7 ^Ad^	27.6 ^CDc^	33.7 ^Cb^	43.7 ^Bca^	0.24	<0.001
		0.05%	20.6 ^d^	25.6 ^Ac^	29.7 ^Bb^	31.9 ^Db^	42.1 ^Ca^	0.18	<0.001
	*p*-value		<0.697	<0.014	<0.001	<0.004	<0.001		

^A–F^ Means in the same column with different superscript of capital letters differ significantly (*p* < 0.05); ^a–e^ Means in the same row with different superscript of small letters differ significantly (*p* < 0.05). Ow: overwrapped, MAP1: 80% O_2_ + 20% CO_2_, MAP2: 40% O_2_ + 30% CO_2_ + 30% N_2_.

**Table 4 foods-12-02916-t004:** Effect of adding cinnamon bark oil on the TBARS (mg MDA/kg) of ground lamb stored at 4 °C for 16 days under different packaging conditions (n = 5).

Treatments		Storage Time (Day)						
Packaging Types	CBO	0	4	8	12	16	SEM	*p*-Value
Ow	0%	0.36 ^d^	0.44 ^Ad^	0.92 ^Ac^	1.65 ^Ab^	2.26 ^Aa^	0.09	<0.001
MAP1	0%	0.35 ^d^	0.40 ^Bd^	0.61 ^Bc^	0.79 ^Bb^	1.16 ^Ba^	0.01	<0.001
	0.025%	0.33 ^d^	0.39 ^Bd^	0.51 ^CDc^	0.75 ^Bb^	0.93 ^CDa^	0.03	<0.001
	0.05%	0.34 ^d^	0.41 ^ABd^	0.58 ^Bc^	0.81 ^Bb^	0.97 ^Ca^	0.11	<0.001
MAP2	0%	0.34 ^c^	0.38 ^Bd^	0.54 ^BCc^	0.63 ^BCb^	0.85 ^DEa^	0.19	<0.001
	0.025%	0.35 ^d^	0.39 ^Bd^	0.46 ^Cc^	0.54 ^Cb^	0.71 ^EFa^	0.05	<0.001
	0.05%	0.36 ^d^	0.39 ^Bd^	0.49 ^CDc^	0.57 ^Cb^	0.75 ^Fa^	0.07	<0.001
*p*-value		<0.082	<0.034	<0.001	<0.001	<0.001		

^A–F^ Means in the same column with different superscript of capital letters differ significantly (*p* < 0.05); ^a–d^ Means in the same row with different superscript of small letters differ significantly (*p* < 0.05). Ow: overwrapped, MAP1: 80% O_2_ + 20% CO_2_, MAP2: 40% O_2_ + 30% CO_2_ + 30% N_2_.

## Data Availability

Data are available from the corresponding author upon request.

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
