# Peer review of "Combined Effect of Cinnamon Bark Oil and Packaging Methods on Quality of Fresh Lamb Meat Patties during Storage at 4 °C"

_foods, 2023, doi:10.3390/foods12152916_

Round 1

Reviewer 1 Report

The manuscript contains results on combined effect of cinnamon bark oil and packaging methods on quality of fresh lamb meat patties during storage at 4 °C. The similar studies have been already conducted by the same group of authors (Hussain, Z., Li, X., Zhang, D., Hou, C., Ijaz, M., Bai, Y., ... & Zheng, X. (2021). Influence of adding cinnamon bark oil on meat quality of ground lamb during storage at 4 C. Meat Science, 171, 108269).The published paper is properly cited and the obtained results are compared. I noted some similarities (introduction, sampling, conclusion...) between these two papers. I don’t have access to read whole manuscript published in Meat Science, so I am not able to compare them in total, but in the present manuscript emphasise is on combined effects of cinnamon bark oil and packaging methods, and in previous only effect of different concentrations of cinnamon bark oil was examined. I think that in discussion part one paragraph should be added regarding this combined effect and interaction of addition of essential oils and packaging methods because that is the main novelty of the present work and in my opinion it is not enough discussed. Also, in introduction part there is nothing about interactions and combined effects of oil addition and packaging methods. All in all there is some novelty in the present paper.

Reviewer 2 Report

The article investigates the effects of incorporating CBO (cinnamon bark oleoresin) along with various packaging methods on the quality of ground lamb meat. The study found that CBO can effectively retard microbial growth in both modified atmosphere packaging (MAP) conditions. The samples stored under MAP2 condition with 0.05% CBO had a better protective effect on inhibiting the populations of TVC, lactic acid bacteria, enterobacteriaceae and pseudomonas during storage.

The study has several limitations. First, the study only used one type of meat (ground lamb). It is not clear if the results would be the same for other types of meat. Second, the study only used two concentrations of CBO (0.025% and 0.05%). It is not clear if the results would be the same for other concentrations of CBO. Moreover, the study only evaluated the effects of CBO on color, oxidation, and microbial quality. It would be interesting to see how CBO affects other quality attributes of ground lamb meat, such as taste and texture.

Overall, the study provides some evidence that CBO can be an effective preservative agent for ground lamb meat. However, more research is needed to confirm these findings and to determine the optimal concentration of CBO for use in ground lamb meat.

See the attachment for more comments.

Reviewer 3 Report

The manuscript by Hussain et al has some significance as it applied various packaging and essential oils to control the oxidative stability and microbial quality of lamb. The language is clear and easy to understand. The hypothesis is sound and may be further improved by incorporating the justification of the concentration of gases in MAL and levels of CBO.

         i.          L19-20: plz mention the method of application

       ii.          L 22: Please mention the storage duration and storage temperature

     iii.          L 22-26: Please also mention the significance level of the results

     iv.          Keywords: Appropriate

       v.          In the methodology, please mention the during of storage

     vi.          L85: plz mention the overwrapping material and specification

   vii.          Authors may put some justification for taking the MAP and the concentration of gases also for improving the hypothesis. For lamb, red meat a higher oxygen concentration is desired and to prevent oxidative changes essential oils were used in the study. Also please reason behind these 0.025 and 0.05% levels (previous literatures or preliminary trials etc). This will strengthen the hypothesis.

  viii.          L84-85: As authors mentioned 7 treatments? Please mention with more clarity. As in abstract L21 mention 3 packaging methods? (It is 3 treatment for MAP1; 3 for MAP 2 +OW)

     ix.          Table 4: the TBARs value on 16 days was higher in 0.050% group as compared to 0.025%

       x.          L322: Please check for vacuum packaging. Authors used MAP

     xi.          L383: Plz mention the maximum permissible limit for TBARS

Round 2

Reviewer 1 Report

Thank you, In my opinion the manuscript is significantly improved during revision so it is suitable for publication. Thank you for your answers.